# INO10, a Chaga Mushroom Extract, Alleviates Alzheimer’s Disease-Related Pathology and Cognitive Deficits in 3xTg-AD Mice

**DOI:** 10.3390/ijms26104729

**Published:** 2025-05-15

**Authors:** Soyoung Ban, Thuong Thi Do, Jang-Won Pyo, Minho Moon, Jong-Tae Park

**Affiliations:** 1Department of Food Science and Technology, Chungnam National University, Daejeon 34134, Republic of Korea; syban@carboexpert.com (S.B.); dothithuong@carboexpert.com (T.T.D.); pyojw0906@naver.com (J.-W.P.); 2CARBOEXPERT Inc., Daejeon 34134, Republic of Korea; 3Department of Biochemistry, College of Medicine, Konyang University, Daejeon 35365, Republic of Korea; 4Research Institute for Dementia Science, Konyang University, Daejeon 35365, Republic of Korea

**Keywords:** Alzheimer’s disease, chaga extract, inotodiol, neuroinflammation, 3xTg mouse

## Abstract

Alzheimer’s disease (AD) is a progressive neurodegenerative disorder characterized by cognitive impairment with amyloid-β (Aβ) accumulation, tau hyperphosphorylation, and neuroinflammation. Among these pathological features, microglial activation is hallmark of neuroinflammation. Chaga (*Inonotus obliquus*) extract has been traditionally used for its diverse pharmacological properties, including anti-inflammatory and neuroprotective effects. This study aimed to evaluate the therapeutic potential of INO10, an inotodiol-rich chaga extract, in murine BV2 microglial cells and a 3xTg-AD mouse model. In BV2 cells, INO10 significantly reduced LPS-induced expression of pro-inflammatory cytokines (IL-1β, IL-6, TNF-α), indicating its potent anti-inflammatory effects. Oral administration of INO10 significantly improved spatial memory in 3xTg-AD mice, as evidenced by increased spontaneous alternation in the Y-maze test. Furthermore, INO10 treatment attenuated neuroinflammation, as indicated by reduced microglial activation and downregulated expression of pro-inflammatory cytokines. In addition, immunohistochemical analysis confirmed that INO10 exhibited favorable bioavailability, supporting its potential as a neuroprotective agent. Histological analysis further revealed a reduction in Ab accumulation and tau phosphorylation in the hippocampus, accompanied by a marked decrease in neuroinflammatory markers. These findings suggest that INO10 effectively mitigates AD-related pathology by reducing Aβ deposition, tau hyperphosphorylation, and neuroinflammation, ultimately leading to cognitive enhancement. Given its multi-target neuroprotective properties, INO10 may serve as a promising natural compound for AD treatment. Further investigations are warranted to elucidate its precise mechanisms and clinical applicability.

## 1. Introduction

Alzheimer’s disease (AD) is a progressive neurodegenerative disorder characterized by cognitive decline, amyloid-β (Aβ) plaque accumulation, tau hyperphosphorylation, Ca^2+^ homeostasis dysregulation, and neuroinflammation [1,2]. In addition to these hallmarks, mitochondrial dysfunction, oxidative stress, and microglial activation are also critical contributors to AD pathology [3,4]. Despite extensive research efforts, current pharmacological interventions for AD remain largely symptomatic and fail to effectively modify disease progression. Moreover, many available drugs are associated with adverse side effects, underscoring the need for alternative therapeutic strategies, particularly those targeting neuroinflammation [5,6]. Given the role of neuroinflammation in AD progression, natural compounds with anti-inflammatory and neuroprotective properties have garnered attention as potential therapeutic candidates.

Chaga (*Inonotus obliquus*), a medicinal fungus, has long been used for its diverse pharmacological activities, including antioxidant, anti-cancer, and immunomodulatory effects [7,8,9]. Traditionally, chaga has been consumed as a tea or decoction by boiling the sclerotium to extract its bioactive compounds [10]. Recent studies have highlighted its potential neuroprotective properties, particularly in modulating neuroinflammation [11,12,13]. Among the major bioactive compounds found in chaga, inotodiol and betulinic acid exhibit strong anti-inflammatory activity by downregulating pro-inflammatory cytokines such as TNF-α, IL-6, and IL-1β [14,15]. While the anti-inflammatory and antioxidant properties of chaga extract have been well documented, the neuroprotective potential of inotodiol-rich chaga extract (INO10) in the context of AD remains largely unexplored [16].

To address this gap, this study aimed to evaluate the therapeutic effects of INO10 in both in vivo and in vitro models of AD. Therefore, in this study, we aimed to evaluate the therapeutic effects of INO10 in both in vivo and in vitro models of AD. Behavioral assessments revealed that INO10 administration significantly improved spontaneous alternation behavior in the Y-maze test. Furthermore, histological analysis revealed a reduction in Aβ accumulation and tau phosphorylation in the hippocampus, alongside decreased microglial activation. These findings suggest that INO10 mitigates AD pathology by modulating neuroinflammation and may serve as a promising natural compound for AD treatment.

Among these compounds, inotodiol has demonstrated significant anti-inflammatory activity by downregulating pro-inflammatory cytokines even at low concentrations [17]. Similarly, betulinic acid exhibits diverse biological activities, including anti-inflammatory, anti-cancer, and anti-viral properties [15]. Given its potential to modulate neuroinflammation, INO10 warrants further investigation as a therapeutic agent for AD and other neuroinflammatory disorders.

Preclinical safety assessments have confirmed that INO20, an extract derived from *Inonotus obliquus*, is non-toxic at doses up to 30–120 mg/kg in mice, with no significant hematological or biochemical abnormalities observed following long-term administration [18,19]. Blood chemistry analysis showed no significant changes in liver function markers (ALT, AST, ALP, total bilirubin) or kidney function markers (BUN, creatine), further supporting a favorable systemic safety profile [18]. Given the compositional similarities between INO10 and INO20, these findings suggest that INO10 may also exhibit a comparable safety profile.

In addition, we aimed to evaluate the neuroprotective effects of INO10 in both in vivo and in vitro models of AD. Specifically, we assessed its impact on cognitive function in 3xTg-AD mice, quantified Aβ accumulation and tau phosphorylation via immunohistochemistry (IHC), and analyzed its anti-inflammatory effects in murine BV2 microglial cells. By elucidating the molecular mechanisms underlying chaga’s neuroprotective properties, this study seeks to provide a scientific basis for its potential application in AD treatment and contribute to the development of novel natural product-based therapeutics for neurodegenerative disease.

## 2. Results

### 2.1. Identification and Quantification of Inotodiol in Chaga Extracts

The HPLC chromatograms demonstrate the identification and quantification of inotodiol in different samples. In Figure 1A, which presents the inotodiol standard, a single sharp peak was detected at 10.913 min, confirming the high purity of the reference compound. Figure 1B, which corresponds to the sample extract, exhibits multiple peaks, with the inotodiol peak appearing at 10.900 min. These results confirm the successful detection of inotodiol and highlight the presence of additional co-extracted components in the sample. Although INO10 contains co-extracted bioactive compounds in addition to inotodiol, we have previously demonstrated the independent neuroprotective efficacy of purified inotodiol-derived compound CE9A215 in 3xTg-AD mice [20], validating the therapeutic role of inotodiol. The current study therefore aimed to evaluate the potential synergistic or additive effects of inotodiol within a natural extract matrix, which better represents real-world application. The difference in chromatographic profiles highlights the complexity of the extract compared to the purified standard. Appendix A further describes the basic composition of INO10/γ-CD, including protein and fat contents below 1%. The mobile phase gradient and instrument setting for HPLC-ELSD analysis, which enabled the detection of inotodiol, are detailed in Appendix A.

### 2.2. Cytotoxicity and Anti-Inflammatory Effects of INO10 in BV2 Microglial Cells

To evaluate the potential cytotoxic effects of INO10, an MTT assay was performed on BV2 microglial cells treated with increasing concentrations of INO10 (0–100 μg/mL) for 24 h (Figure 2A). The results indicated that INO10 did not induce significant cytotoxicity at lower concentrations. However, at concentrations ≥ 50 μg/mL, a marked reduction in cell viability was observed, suggesting potential cytotoxic effects at higher doses.

To assess the anti-inflammatory properties of INO10, we examined the mRNA expression levels of key pro-inflammatory cytokines (*IL-1β*, *IL-6*, and *TNF-α*) in LPS-stimulated BV2 cells. LPS (0.1 μg/mL) significantly upregulated *IL-1β* mRNA expression compared to the control group (*p* < 0.0001) (Figure 2B). However, treatment with INO10 (2.5–20 μg/mL) resulted in a dose-dependent reduction in *IL-1β* expression, with significant suppression observed at doses ≥ 5 μg/mL (*p* < 0.0001), with the highest effect at 20 μg/mL. A similar pattern was observed for *IL-6*, where LPS markedly increased its expression (Figure 2C), and INO10 significantly attenuated *IL-6* levels in a dose-dependent manner. Likewise, *TNF-α* expression was strongly elevated following LPS stimulation (Figure 2D), while INO10 treatment effectively downregulated *TNF-α* expression, with notable effects observed at doses ≥10 μg/mL. To determine whether INO10 modulates the NF-κB signaling pathway, we examined the expression of *Rela* (p65) and *Nfkb1* (p50), two key transcription factors involved in inflammation. LPS stimulation resulted in a slight increase in *Rela* mRNA expression, though it was not statistically significant. INO10 treatment did not significantly alter *Rela* expression at the tested concentrations (Figure 2E). Similarly, *Nfκb1* expression remained unchanged following LPS stimulation and INO10 treatment (Figure 2F), suggesting that the observed anti-inflammatory effects of INO10 are independent of direct modulation of *NF-kB* transcription factor expression. A comprehensive summary of INO10-induced changes in the mRNA expression of pro-inflammatory cytokine (*TNF-α*, *IL-6*, *IL-1β*) and NF-κB-related transcription factors (*Nfkb* and *Rela*) is provided in Appendix A.

### 2.3. Pharmacokinetic Profile of INO10 Formulations

The pharmacokinetic properties of INO10 were evaluated using three different formulations: INO10/γCD_5% Kolliphor, INO10 in 5% Kolliphor, and INO10_5% HPMC. Plasma concentration–time profiles of inotodiol are presented in Figure 3A–C. INO10/γCD_KP (Figure 3A) exhibited the highest Cmax (36.7 ng/mL), indicating that γ-cyclodextrin (γCD) significantly enhances INO10 solubility and absorption. In comparison, INO10_KP showed a moderate Cmax (21.5 ng/mL), suggesting that Kolliphor (KP) alone improves bioavailability but is less effective than γCD complexation. Meanwhile, INO10_HM displayed the lowest Cmax (7.8 ng/mL), implying poor absorption and limited bioavailability. These results suggest that γCD complexation enhances INO10’s pharmacokinetic properties by increasing its solubility and gastrointestinal absorption, making it a promising formulation strategy for optimizing absorption efficiency.

### 2.4. Effect of INO10 on Cognitive Impairment in 3xTg Mice

To investigate the effects of INO10 on cognitive deficits, we performed behavioral assessments in 3xTg-AD mice. The overall experimental design, including the treatment schedule and timing of behavioral and histological analysis, is illustrated in Appendix A. Spatial memory was evaluated using the Y-maze test, where INO10-treated 3xTg mice exhibited significantly increased spontaneous alternations compared to vehicle-treated 3xTg-AD mice (Figure 4A). In contrast, the total number of arm entries did not differ between the groups (Figure 4B), suggesting that INO10 does not affect general locomotor activity. These results indicate that INO10 effectively improves cognitive dysfunction in an animal model of Alzheimer’s disease (AD).

We further examined its effects on AD-related pathology, including Aβ accumulation, tau hyperphosphorylation, and neuroinflammation, in 3xTg-AD mice. Immunohistochemical analysis using the 4G8 antibody revealed a significant reduction in Aβ accumulation in both the subiculum and the stratum pyramidale (SP) of CA1 in INO10-treated 3xTg mice compared to vehicle-treated controls (Figure 4A–C). Further sex-specific analysis demonstrated that INO10 effectively reduced Aβ pathology in both male and female 3xTg mice (Appendix A).

To assess the impact of INO10 on tau pathology, immunofluorescence staining with the AT8 antibody was performed, revealing a significant decrease in tau hyperphosphorylation at Ser202/Thr205 in the subiculum and SP of CA1 in INO10-treated mice compared to controls (Figure 4A–C). Similar to Aβ pathology, INO10 attenuated tau hyperphosphorylation in both male and female 3xTg mice (Appendix A).

Furthermore, we investigated the effects of INO10 on neuroinflammation using Iba-1 immunofluorescence staining, a marker of microglia activation. A marked reduction in microgliosis was observed in the subiculum and SP of CA1 in INO10-administered mice, indicating significant attenuation of neuroinflammation (Figure 4A–C). Notably, these anti-inflammatory effects were evident in both male and female 3xTg mice (Appendix A). Collectively, these findings suggest that INO10 administration effectively mitigates Aβ accumulation, tau hyperphosphorylation, and neuroinflammation in 3xTg-AD mice, highlighting its potential as a promising therapeutic candidate for AD.

## 3. Discussion

Neuroinflammation, driven by microglial activation and excessive pro-inflammatory cytokine release, is a key contributor to Alzheimer’s disease (AD) pathology, with the NF-κB signaling pathway playing a central role in sustaining chronic inflammation. Our study demonstrated that INO10, a chaga extract containing 10% inotodiol, exerts strong anti-inflammatory effects (Figure 2B–D). However, despite this response, INO10 did not alter NF-κB subunit mRNA levels (Figure 2E,F), suggesting that its mechanism of action may involve post-translational regulation or alternative inflammatory pathways rather than direct NF-κB transcriptional suppression [21]. Given that NF-κB activation is primarily regulated by phosphorylation and nuclear translocation, the fact that the mRNA levels of its downstream targets remained unchanged suggests that INO10 may exert its effects at the post-translational level by preventing NF-κB phosphorylation, thereby blocking its activation. These findings suggest that INO10’s anti-inflammatory effects may be mediated through post-translational modifications rather than direct suppression of NF-κB transcription.

Pharmacokinetic analysis demonstrated that INO10/γCD_KP (Figure 3A) exhibited the highest Cmax (36.7 ng/mL), while INO10_KP (Figure 3B) and INO10_HM (Figure 3C) showed lower Cmax values of 21.5 ng/mL and 7.8 ng/mL, respectively. These findings align with previous studies demonstrating that γ-CD improves the solubility and gastrointestinal absorption of poorly water-soluble compounds by forming an inclusion complex, thereby enhancing their bioavailability and therapeutic efficacy [22]. Notably, the KP-only formulation resulted in improved bioavailability compared to the HPMC formulation but was less effective than γ-CD complexation. This suggests that while Kolliphor enhances solubility and absorption to some extent, γ-CD complexation provides a more significant improvement in pharmacokinetic performance [23]. The difference between the γ-CD and KP formulation indicates that γ-CD likely plays a crucial role in stabilizing and enhancing the solubility of INO10 in the gastrointestinal tract, facilitating better absorption.

INO10 administration effectively alleviates cognitive deficits and reduces AD-related pathology in 3xTg-AD mice. Behavioral assessments revealed that INO10 significantly improved spatial memory, as evidenced by increased spontaneous alternations in the Y-maze test, without affecting locomotor activity (Figure 4B,C). These findings suggest that INO10 enhances cognitive function independently of general motor performance.

At the molecular level, INO10 treatment led to a significant reduction in Aβ accumulation in the subiculum and stratum pyramidale of CA1, as indicated by immunohistochemical analysis (Figure 4D,G). These results align with previous studies demonstrating that natural bioactive compounds can attenuate Aβ deposition through enhanced clearance mechanisms or reduced Aβ production. INO10 is a chaga extract containing 10% inotodiol, a bioactive triterpenoid known for its anti-inflammatory and anti-allergic properties [24]. Inotodiol has been traditionally used in herbal medicine and has been reported to be non-toxic at doses up to 20 mg/kg [25]. Given its pharmacological profile, inotodiol may contribute to the observed neuroprotective effects of INO10 by modulating inflammatory responses and promoting Aβ clearance [26,27]. In addition to inotodiol, INO10 contains other bioactive components that may exert synergistic effects. Betulin, a naturally occurring pentacyclic triterpenoid found in chaga, shares compositional similarities with INO20 and has been shown to exhibit diverse biological activities, including anti-inflammatory, anti-viral, and anti-cancer effects [7,18,28]. Previous studies suggest that betulin plays a role in suppressing pro-inflammatory cytokines, indicating that it may act in concert with inotodiol to modulate neuroinflammation [15]. The combined effects of these components suggest that INO10’s therapeutic potential is likely mediated by multiple bioactive compounds working together rather than a single active ingredient.

In addition to Aβ pathology, INO10 significantly reduced tau hyperphosphorylation at Ser202/Thr205 in the hippocampal regions of 3xTg-AD mice (Figure 4E,H). Tau hyperphosphorylation is a hallmark of AD that contributes to the formation of neurofibrillary tangles, ultimately leading to neuronal dysfunction and cognitive decline [29]. The ability of INO10 to attenuate tau phosphorylation suggests that it may interfere with kinases such as GSK-3β or CDK5, which are known to regulate tau phosphorylation [20]. Further studies are warranted to elucidate the precise molecular mechanism by which INO10 modulates tau pathology. Neuroinflammation plays a crucial role in AD progression, and excessive activation of microglia contributes to disease pathology [30]. In this study, we observed a significant reduction in microglia in INO10-treated mice, as evidenced by decreased Iba-1 expression (Figure 4I,F). These findings suggest that INO10 exerts anti-inflammatory effects, potentially by modulating the NF-κB signaling pathway or suppressing the release of pro-inflammatory cytokines such as IL-1β, IL-6, and TNF-α. The reduction in neuroinflammation observed in both male and female 3xTg-AD mice further supports the therapeutic potential of INO10 in AD.

Although our findings provide compelling evidence for the neuroprotective effects of INO10, this study has several limitations. First, while our data suggest that INO10 reduces Aβ accumulation, tau hyperphosphorylation, and neuroinflammation, the precise molecular mechanisms underlying these effects remain unclear. Given that INO10 did not alter NF-κB subunit mRNA expression in BV2 microglia cells but still exhibited anti-inflammatory effects, its regulatory mechanisms may involve post-transcriptional modifications or alternative signaling pathways such as Nrf2/HO-1. Future studies employing transcriptomic and proteomic analyses may provide deeper insights into INO10’s mode of action. Second, our study primarily focused on the hippocampus, yet AD pathology extends to other brain regions such as the cortex. Investigating the effects of INO10 on these regions could provide a more comprehensive understanding of its therapeutic potential. Moreover, the differences observed between in vitro and in vivo models highlight the need for further studies to explore the cumulative effects of long-term INO10 administration and its potential interactions with other cell types in the brain microenvironment. Lastly, while INO10 showed promising effects in a transgenic mouse model, further validation in other AD models and clinical studies is necessary to confirm its translational relevance.

## 4. Materials and Methods

### 4.1. Materials

Dulbecco’s Modified Eagle’s medium (DMEM), Dulbecco’s Phosphate Buffered Saline (DPBS), fetal bovine serum (FBS) and penicillin–streptomycin (P/S) were purchased from Sigma-Aldrich (St. Louis, MO, USA). 3-(4,5-Dimethylthiazol-2yl)-2,5-diphenyltetrazolium bromide (MTT) was obtained from Thermo Scientific (Waltham, MA, USA). For RNA extraction and qPCR analysis, TRIzol reagent (Life Technologies, Carlsbad, CA, USA), AccuPower CycleScript RT Premix, and AccuPower Taq PCR Premix were used. Primers were purchased from Bioneer (Daejeon, Republic of Korea).

### 4.2. Preparation of INO10

Chaga mushroom powder was obtained from Jungwoodang Company (Seoul, Republic of Korea). The purification and purity verification of CE9A215 were performed as previously described [18]. Briefly, 200 g of chaga mushroom powder was extracted with 2 L of 70% ethanol and stirred at 50 °C for 4 h. The extract was centrifuged at 12,000× *g* for 10 min at room temperature, and the resulting supernatant was concentrated using a vacuum rotary evaporator (Daehan Science Co., Daejeon, Republic of Korea) to a final volume of 200 mL. The concentrate was then stored at 4 °C for 15 h, and the precipitant was separated by centrifugation. The precipitate obtained at this stage was designated as inotodiol concentrate (INO10), which contained 10% inotodiol. Further purification of INO10 was conducted using a recycling high-performance liquid chromatography (HPLC) system (MPLC, YMCKOREA, Seongnam, Republic of Korea) equipped with reverse-phase chromatography (ODS-AQ, 50 × 500 mm).

### 4.3. Preparation of INO10/γ-CD Mixture

The inclusion complex was prepared following the protocol described in a previous studies [31]. The INO10 and γ-cyclodextrin (γ-CD) mixture was prepared at a 1:4 weight ratio (*w*/*w*). Specifically, 60 mg of INO10 was combined with 240 mg of γ-CD, ensuring a total weight ratio of 20% INO10 and 80% γ-CD. The two components were thoroughly mixed using a probe ultrasonic processor (Q500, Qsonica, Newtown, CT, USA) equipped with a microtip at 30% amplitude for 10 min, followed by a horn tip at 50% amplitude for 5 min. Dried powder samples were stored at 4 °C avoiding exposure to light prior to use.

### 4.4. Cell Culture

BV2 cells, a murine microglial cell line, were kindly provide by Dr. Jongil Park from Chungnam University Medical School. The cells were maintained in Dulbecco’s Modified Eagle’s Medium (DMEM) supplemented with 10% fetal bovine serum (FBS) and 1× penicillin–streptomycin. BV2 cells were cultured at 37 °C in a humidified incubator with 5% CO_2_.

### 4.5. Cell Viability Assay

A standard MTT assay was performed to assess the cytotoxic effects of INO10. BV2 cells were seeded at a density of 1 × 10^5^ cells per well in a 96-well plate and incubated for 24 h. Cells were then treated with various concentrations of INO10 (0~100 μg/mL) for an additional 24 h under 5% CO_2_. Following treatment, 3-(4,5-dimethylthiazol-2yl)-2,5-diphenyltetrazolium bromide (MTT) solution was added to each well at a final concentration of 5 mg/mL and incubated for 2 h. The resulting formazan crystals were dissolved in 100 μL of dimethyl sulfoxide (DMSO), and absorbance was measured at 570 nm using a microplate reader. Cell viability was expressed as percentage relative to the untreated control group.

### 4.6. RNA Extraction and Quantitative PCR Analysis

BV2 cells were seeded in 6-well plates and pretreated with INO10 at a final concentration of 0, 2.5, 5, 10, and 20 μg/mL for 24 h, followed by LPS stimulation (0.1 μg/mL) for 4 h. RNA extraction, cDNA synthesis, and PCR amplifications were performed using the reagents described in the Materials section. Specific primers were used in qPCR analysis for *TNF-α*, *IL-1β*, *IL-6*, *Rela* (p65), and *Nfkb1* (p50), with GAPDH as an internal control. Primer sequences are listed in Appendix A.

### 4.7. Animals

To determine the concentration of inotodiol in blood, pharmacokinetic analysis was conducted using ICR mice. Eight 8-week-old male ICR mice were purchased from SAMTAKO Bio Korea (Osan, Republic of Korea) and randomly assigned into two groups (n = 3 per group).

To further investigate the therapeutic effects of INO10 in AD, triple transgenic (3xTg) mice were used. This model carries three familial AD mutations: human PSEN1 (M146V), human APP (Swedish K670N/M671L), and human MAPT (P301L), and was obtained from Jackson Laboratory (Strain #004807, Bar harbor, ME, USA). For the experiment, 7-month-old 3xTg-AD mice were randomly assigned to two groups: (1) vehicle-treated 3xTg-AD mice group (n = 15), (2) INO10-treated 3xTg-AD group (5 mg/kg, n = 15). Vehicle or INO10 (5 mg/kg) was administered five times per week (Monday–Friday) for 4 months (Appendix A). Seven-month-old 3xTg-AD mice were selected because, at this age, they begin to exhibit both amyloid-β deposition and early tau hyperphosphorylation, along with initial cognitive impairment, making it a suitable stage to evaluate therapeutic interventions targeting early pathological changes in Alzheimer’s disease.

All animals were housed under standard laboratory conditions, with a temperature maintained at 20–25 °C, relative humidity of 30–35%, and a 12 h light/dark cycle. The mice were provided with a standard diet and allowed to acclimate prior to experimentation. All procedures were conducted in accordance with ethical guidelines and were approved by the Institutional Animal Care and Use Committee (IACUC) of Chungnam National University (202103A-CNU-076).

### 4.8. Oral Bioavailability in Mice

Following a 3 h fasting period, test solutions of INO10 dissolved in 5% Kolliphor HS 15 (BASF, Ludwigshafen, Germany) and 5% Hydroxy Propyl Methyl Cellulose (HPMC). These solutions were orally administered at a dose of 0.5 mg/kg (INO10 group). Blood samples were collected from the tail vein at predefined time points post-administration and transferred into heparinized plasma tubes. Plasma was separated by centrifugation at 4000 rpm for 10 min and stored at –80 °C until further analysis. The concentration of inotodiol in plasma was quantified using ultra-high-performance liquid chromatography coupled with tandem mass spectrometry (UHPLC/MS/MS; 1290 Infinity, Agilent Technologies, Santa Clara, CA, USA). Chromatographic separation was conducted on a Zorbax Eclipse Rapid Resolution High Definition C18 column (2.1 × 100 mm, 1.8 μm, Agilent Technologies) using a gradient elution system.

For mobile phases composition, solution A (distilled water) and solution B (methanol) were both supplemented with 5 mM ammonium acetate and 0.1% formic acid. The mass spectrometer operated in positive ion mode, employing a mobile phase gradient as detailed in Appendix A. The purity of inotodiol was verified via scan mode within a mass range of 300 to 1000 *m*/*z*, and its plasma concentration was determined using multiple reaction monitoring (MRM) mode. A standard curve was generated over a range of 1–100 ppb, with the limits of detection (LOD) and quantitation (LOQ) set at 1 ppb and 3 ppb, respectively. All samples were analyzed in triplicate within each experiment, and the entire experiment was independently repeated at least three times.

### 4.9. Behavioral Test

To assess short-term spatial recognition memory, the Y-maze test was conducted. The experimental environment was maintained at 23 ± 1 °C with 60 ± 10% humidity. The Y-maze consisted of three arms (each 8 cm wide × 30 cm long × 15 cm high) arranged at 120° angles. Prior to the behavioral test, the mice underwent a one-week adaptation period to minimize stress and ensure reliable memory function assessment. During this period, each mouse was placed in a dark environment under the same experimental conditions for one hour per day. Mice were then allowed to explore the Y-maze freely for 5 min to familiarize themselves with the apparatus. Following the final administration of vehicle or INO10, the Y-maze test was conducted. Each mouse was placed in the center of the maze and allowed to freely explore the three arms for 8 min. Total arm entries and spontaneous alternations were recorded manually. Spontaneous alternations were defined as three consecutive entries into different arms without revisiting a previous arm (e.g., ABC, BAC, or CBA), while repeated entries were not counted (e.g., ABA or CBC). The spontaneous alternation percentage was calculated using the following formula: [(the number of alterations)/(the total number of arms entered − 2)] × 100, where the subtraction accounts for the first two entries that do not constitute a full alternation.

### 4.10. Preparation of Brain Tissue

For histological analysis, vehicle- or INO10-administrated mice were sacrificed. Animals were anesthetized via intraperitoneal injection of Avertin (Tribromoethanol; Sigma-Aldrich, St. Louis, MO, USA) at a dose of 250 μg/kg. Cardiac-perfusion was sequentially performed with 0.05 M phosphate-buffered saline (PBS) followed by 4% paraformaldehyde (PFA). Extracted brains were post-fixed in 4% PFA at 4 °C for 2 h, then transferred to a 30% sucrose solution for cryoprotection. Brains were then sectioned into 30 μm-thick coronal sections at −25 °C using a CM1850 cryostat (Leica Biosystems, Wetzlar, Germany).

### 4.11. Immunofluorescence Labeling

To assess immunoreactivity of markers associated with AD pathology, two to three coronal brain sections were obtained at the level of the dorsal subiculum (−2.80 and −3.80 mm to the bregma) and dorsal CA1 (−1.58 and −2.18 mm to the bregma). The sections were briefly rinsed in PBS containing 0.5 mg/mL bovine serum albumin and 0.3% Triton X-100, followed by overnight incubation at 4 °C with the following primary antibodies: mouse anti-4G8 antibody (1:2000; BioLegend, San Diego, CA, USA), mouse anti-phospho-tau (AT8) antibody (1:200; Thermo Fisher Scientific Inc., Waltham, MA, USA), goat anti-ionized calcium-binding adapter molecule 1 (Iba-1) antibody (1:500; Abcam, Cambridge, MA, USA). For 4G8 staining, sections were incubated with 70% formic acid for 20 min for antigen retrieval. After primary antibody incubation, sections were washed and incubated for 1 h at room temperature with the following secondary antibodies (1:200; Thermo Fisher Scientific): Donkey Alexa 594-conjugated anti-rabbit IgG, Donkey Alexa Fluor Plus 488-conjugated anti-goat IgG, and Donkey Alexa Fluor Plus 488-conjugated anti-mouse IgG (1:200; Thermo Fisher Scientific). Nuclei w 4,6-diamidino-2-phenylindole (DAPI) staining and mounting was performed using Fluoroshield™ with DAPI (Sigma-Aldrich, St. Louis, MO, USA).

### 4.12. Image Acquisition and Analysis

Images of brain tissue sections were acquired using a Zeiss LSM 700 (Carl Zeiss AG, Oberkochen, Germany). Images analysis was conducted using ImageJ software (NIH, Bethesda, MD, USA; https://imagej.net/ij/, accessed on 15 December 2023). The immunoreactivity of 4G8, AT8, and Iba-1 was quantified by measuring the area fractions of immune-positive signals in the brain tissues. All image acquisition and analysis were performed in a blinded manner to minimize bias.

### 4.13. Statistical Analysis

All statistical analyses were performed using GraphPad Prism 10.0 software (GraphPad Software, La Jolla, CA, USA). Data are presented as mean ± standard error of the mean (SEM). Statistical comparisons between three or more groups were conducted using an independent *t*-test. Differences were considered statistically significant at *p* < 0.05.

## 5. Conclusions

In conclusion, our study provides strong evidence that INO10 administration mitigates AD-related pathology and improves cognitive function in 3xTg-AD mice. These findings highlight the potential of INO10 as a promising therapeutic candidate for AD, particularly as a natural product-based intervention rooted in traditional medicine. Given its multi-target pharmacological properties, further investigation into its underlying mechanisms and clinical applicability is warranted.

## Figures and Tables

**Figure 1 ijms-26-04729-f001:**
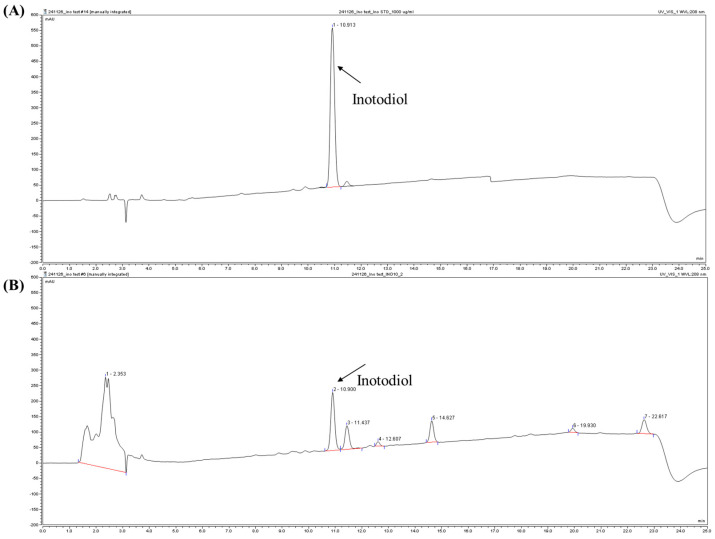
High-performance liquid chromatography (HPLC) chromatograms of inotodiol. Representative HPLC chromatograms of inotodiol standard (**A**) and INO10 from *Inonotus obliquus* (**B**).

**Figure 2 ijms-26-04729-f002:**
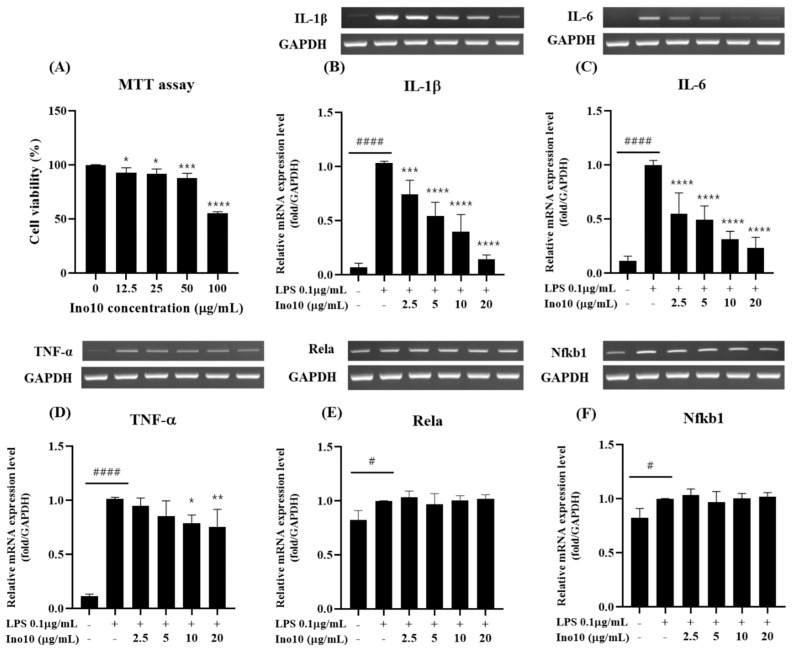
Measurement of INO10 cytotoxicity and establishment of an LPS-induced inflammation model. (**A**) BV2 cell viability was assessed following treatment with increasing concentrations (0–100 μg/mL) of INO10 for 24 h. Cytotoxicity was evaluated using an MTT assay. (**B**–**F**) BV2 cells were treated with LPS (0.1 μg/mL), and the mRNA expression levels of *IL1β* (**B**), *IL-6* (**C**), *TNF-α* (**D**), *Rela* (**E**), and *Nfkb1* (**F**) were analyzed by qRT-PCR. Data are presented as mean ± SD. Statistical significance is indicated as follows: * *p* < 0.05; ** *p* < 0.01; *** *p* < 0.001; **** *p* < 0.0001 vs. the LPS-treated group. # *p* < 0.05, #### *p* < 0.0001 vs. the control group.

**Figure 3 ijms-26-04729-f003:**
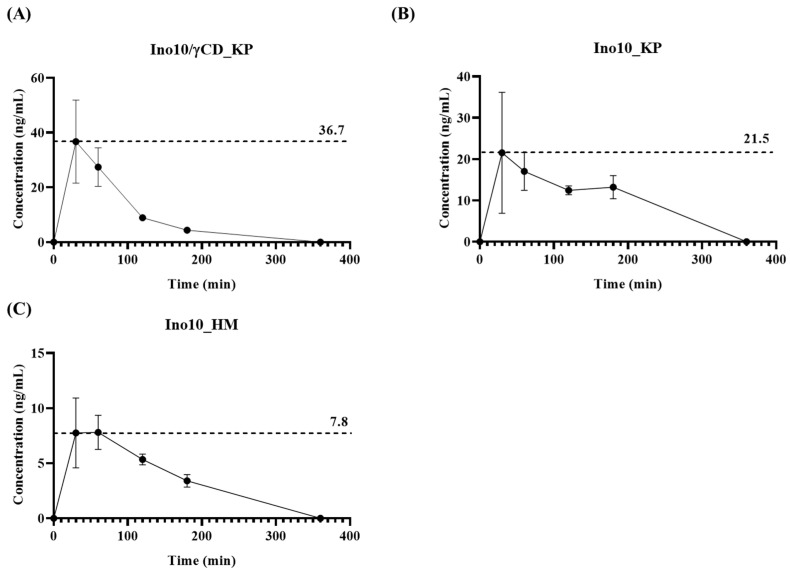
Plasma concentration–time curve of inotodiol following oral administration of different formulations. (**A**) INO10/γCD_5% KP: inclusion complex of 10% inotodiol with γ-CD at a molar ratio of 1:3 in 5% Kolliphor; (**B**) INO10_5% KP, 10% inotodiol in 5% Kolliphor; (**C**) INO10_5% HM: 10% inotodiol in 5% HPMC. Data are presented as mean ± SD (n = 3).

**Figure 4 ijms-26-04729-f004:**
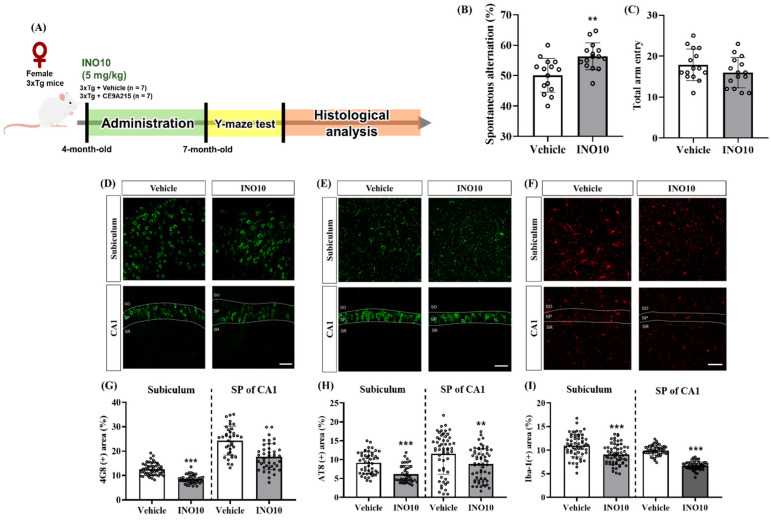
The ameliorative effect of INO10 on cognitive function in 3xTg mice. (**A**) Schematic representation of the experimental design. Female 3xTg mice were orally administered INO10 (5 mg/kg) or vehicle for 3 months from 4 to 7 months of age, followed by the Y-maze test and histological analysis. (**B**) Spontaneous alternation performance was assessed in the Y-maze test 24 h after the final administration. INO10-administered 3xTg mice (n = 15) showed a significant increase in spontaneous alternation percentage compared to vehicle-administered 3xTg mice (n = 15). (**C**) The total number of arm entries during the 8 min test period was not significantly different between the groups. (**D**–**F**) Representative immunofluorescence images of brain sections stained for 4G8 (**D**), AT8 (**E**), and Iba-1 (**F**) in the subiculum and stratum pyramidale (SP) of CA1. Scale bars = 50 μm. (**G**–**I**) Quantification of immunoreactivity for 4G8 (**G**), AT8 (**H**), and Iba-1 (**I**) in the subiculum and SP of CA1. Values are expressed as mean ± SEM. ** *p* < 0.01, *** *p* < 0.001 compared with the vehicle-treated 3xTg mice (black bar).

## Data Availability

Data are contained within the article and Appendix A.

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
