# Peer review of "INO10, a Chaga Mushroom Extract, Alleviates Alzheimer’s Disease-Related Pathology and Cognitive Deficits in 3xTg-AD Mice"

_ijms, 2025, doi:10.3390/ijms26104729_

Round 1
Reviewer 1 Report
Comments and Suggestions for Authors
I found this an interesting and novel study and overall is written well and clearly for the most part. I have a number of generally small issues I think could help the authors improve a few areas where improvements in clarity would help the article in my view.
Abstract:
- the species of the microglial cells studied should be stated
- the mode of administration of INO10 to the 3xTgAD should be stated
lines 25-27 on page 1: there is nothing mentioned earlier in the abstract to support this statement - but on page 2 lines 58-60 the information that backs this up is used and I suggest could be moved to the abstract?
Introduction:
- I think a little more information on how Chaga is ordinarily prepared/taken would be helpful as it will be unfamiliar to many readers
- lines 58-63 page 2 - i think this text is in the wrong place and would ordinarily be used in the concluding paragraph of an introduction - that is followed by other information that might be better introduced before this makes the 'flow' of the narrative disjointed.
- line 81 page 2 - mention the species of the BV2 cells?
Results:
- lines 92-94 page 3 - More information should be provided regarding the co-extracted compounds, was there any attempts, or would it have been possible to do a purified extract from this? Otherwise the study will be criticised that any other co-extracted compound could have elicited the effects observed- especially if there is no way to specifically block inotodiol - this currently a big weakness and would need some further attention.
- line 108 page 3 - is IL-1b a gene symbol? if so then should be in italics - should that be beta? Also - it was unclear in some places whether the authors were referring to genes (which are expressed) and proteins (that technically are not expressed but I appreciate are often described as such). This can be addressed by using italicised text for gene symbols and normal text (abbreviations) for protein symbols.
- lines 152-153 - the use of the word 'controls' is confusing here - do the authors mean it to represented vehicle treated 3xTgAD mice or WT mice ... if the former they could be perhaps described as the control arm because otherwise it might be misinterpreted as control (WT) mice for the 3xTgAD?
Methods:
I think a clear rationale for the selection of the age of the mice when given treatment would be helpful to provide context for what might be the emergence of phenotype and pathology that are relevant to this model and over what timeframe.
Author Response
Comment 1: the species of the microglial cells studied should be stated
Answer: Thank you for your valuable comment. In the revised Abstract, we have clarified that the BV2 cells used in this study are murine-derived microglial cells to provide clearer information regarding the species.
Comment 2: the mode of administration of INO10 to the 3xTgAD should be stated. lines 25-27 on page 1: there is nothing mentioned earlier in the abstract to support this statement - but on page 2 lines 58-60 the information that backs this up is used and I suggest could be moved to the abstract?
Answer: Thank you for your insightful feedback. In the revised Abstract, we have clearly stated that INO10 was orally administered to 3xTg-AD mice. Additionally, to improve the logical flow, we incorporated the relevant supporting information regarding INO10’s bioavailability into the Abstract from the Results section as suggested.
Introduction:
Comment 3: I think a little more information on how Chaga is ordinarily prepared/taken would be helpful as it will be unfamiliar to many readers
Answer: Thank you for the valuable suggestion. In response, we have added a brief description in the introduction section explaining the traditional preparation methods of chaga mushroom. Specifically, we mentioned that Chaga is commonly consumed as a hot water extract (tea) or used in ethanolic extracts for its medicinal properties.
Comment 4: lines 58-63 page 2 - i think this text is in the wrong place and would ordinarily be used in the concluding paragraph of an introduction - that is followed by other information that might be better introduced before this makes the 'flow' of the narrative disjointed.
Answer: Thank you for this helpful comment. We agree that the information regarding the therapeutic effects of INO10 fits more appropriately at the conclusion of the introduction section. Accordingly, we have relocated the sentences describing INO10’s impact on Ab accumulation, tau phosphorylation, and neuroinflammation to the final paragraph of the introduction, thereby improving the logical flow and narrative structure.
Comment 5: line 81 page 2 - mention the species of the BV2 cells?
Answer: Thank you for your insightful comment. We have now specified that BV2 cells are a murine (mouse-derived) microglial cell line in the revised manuscript to provide greater clarity.
Results:
Comment 6: lines 92-94 page 3 - More information should be provided regarding the co-extracted compounds, was there any attempts, or would it have been possible to do a purified extract from this? Otherwise the study will be criticised that any other co-extracted compound could have elicited the effects observed- especially if there is no way to specifically block inotodiol - this currently a big weakness and would need some further attention.
Answer: Thank you for this important comment. We agree that the presence of co-extracted compounds in INO10 could complicate interpretation. To address this concern, we have added clarifications in the revised manuscript. Although INO10 contains multiple bioactive compounds, we have previously demonstrated the independent neuroprotective efficacy of purified inotodiol-derived CE9A215 in 3xTg-AD mice, validating the therapeutic role of inotodiol itself (Ref [20]). Therefore, this study aimed to evaluate the effects of inotodiol in the context of a natural extract matrix, better reflecting real-world application. We have now included a statement regarding the presence of co-extracted compounds and referred to our previous study using purified material to support the therapeutic relevance of inotodiol.
Comment 7: line 108 page 3 - is IL-1b a gene symbol? if so then should be in italics - should that be beta? Also - it was unclear in some places whether the authors were referring to genes (which are expressed) and proteins (that technically are not expressed but I appreciate are often described as such). This can be addressed by using italicised text for gene symbols and normal text (abbreviations) for protein symbols.
Answer: Thank you for your insightful comment regarding gene and protein notation. In the revised manuscript, we have corrected the symbol for interleukin-1 beta from “IL-1b” to “IL-1b” throughout the text. Furthermore, we have clarified the distinction between gene and protein references: gene symbols are now italicized, while protein abbrevations remain in regular font. We have also revised the figure legends accordingly to maintain consistency throughout the manuscript.
Comment 8: lines 152-153 - the use of the word 'controls' is confusing here - do the authors mean it to represented vehicle treated 3xTgAD mice or WT mice ... if the former they could be perhaps described as the control arm because otherwise it might be misinterpreted as control (WT) mice for the 3xTgAD?
Answer: Thank you for pointing out this ambiguity. In the revised manuscript, we have clarified that the term “control group” refers specifically to vehicle-treated 3xTg-AD mice, not to wild-type (WT)mice. To prevent any misunderstanding, we now describe them as the vehicle-treated 3xTg-AD group throughout the text, including the figure legends.
Methods:
Comment 9: I think a clear rationale for the selection of the age of the mice when given treatment would be helpful to provide context for what might be the emergence of phenotype and pathology that are relevant to this model and over what timeframe.
Answer: Thank you for your helpful suggestion. We have now included a clear rationale for the selection of mouse age in the Methods section. Specifically, we selected 7-month-old 3xTg-AD mice because, at this stage, these mice exhibit early pathological hallmarks of Alzheimer’s disease, including amyloid b deposition, tau hyperphosphorylation, and initial cognitive impairments. This time point is widely recognized as a critical period for therapeutic intervention in this model.

Reviewer 2 Report
Comments and Suggestions for Authors
The title is "INO10, a chaga mushroom extract, alleviates Alzheimer’s disease-related pathology and cognitive deficits in 3xTg-AD mice" which explores the potential therapeutic effects of an inotodiol-rich chaga mushroom extract (INO10) on Alzheimer’s disease (AD). The overall structure is clear, but there are some minor issues, suggesting a Minor Revision.
- In HPLC, the differences in chromatograms compared to standard compounds highlight the complexity of the extract versus purified standards. Is it possible that other impurity components may have synergistic or antagonistic effects on subsequent experiments?
- After preparing the INO10/γ-CD mixture, the authors did not confirm the encapsulation efficiency of INO10 by γ-CD. It may be advisable to determine the encapsulation rate of the complex via HPLC or UV spectrophotometry.
Author Response
The title is "INO10, a chaga mushroom extract, alleviates Alzheimer’s disease-related pathology and cognitive deficits in 3xTg-AD mice" which explores the potential therapeutic effects of an inotodiol-rich chaga mushroom extract (INO10) on Alzheimer’s disease (AD). The overall structure is clear, but there are some minor issues, suggesting a Minor Revision.
Comment 1: In HPLC, the differences in chromatograms compared to standard compounds highlight the complexity of the extract versus purified standards. Is it possible that other impurity components may have synergistic or antagonistic effects on subsequent experiments?
Answer: Thank you for your valuable comment. As shown in the HPLC chromatograms, INO10 contains co-extracted compounds in addition to inotodiol, suggesting potential synergistic or antagonistic interactions. However, we have previously demonstrated the independent neuroprotective and anti-inflammatory effects of purified inotodiol-derived CE9A215 in 3xTg-AD mice (Ref [20]), validating the therapeutic relevance of inotodiol itself. Thus, this study aimed to further explore the synergistic or additive potential of inotodiol within a natural extract context, representing a more practical application of Chaga-derived therapeutics.
Comment 2: After preparing the INO10/γ-CD mixture, the authors did not confirm the encapsulation efficiency of INO10 by γ-CD. It may be advisable to determine the encapsulation rate of the complex via HPLC or UV spectrophotometry.
Answer: Thank you for suggestion. In the present study, we did not directly quantify the encapsulation efficiency of INO10 by g-cyclodextrin (g-CD). However, the complexation was performed under standard and optimized conditions, as previously reported for inotodiol-g-CD complexes (Ref [31]). In particular, prior studies demonstrated that g-CD complexation significantly enhanced the bioavailability and anti-inflammatory efficacy of inotodiol (‘Enhancement of bioavailability and anti-inflammatory activity of inotodiol through complexation with g-cyclodextrin’), supporting the validity of this formulation approach. Future work will aim to further characterize encapsulation efficiency by HPLC or UB spectrophotometric methods.

Round 2
Reviewer 1 Report
Comments and Suggestions for Authors
I thank the authors for their consideration of my comments and suggestions and have nothing further to request.